# Highly Dispersible and Stable Carbon Nanotube Ink with Silicone Elastomer for Piezoresistive Sensing

**DOI:** 10.3390/mi17010014

**Published:** 2025-12-24

**Authors:** Hyun Jae Lee, Justin Rejimone, Simon S. Park, Keekyoung Kim

**Affiliations:** 1Department of Mechanical and Manufacturing Engineering, Shulich School of Engineering, University of Calgary, Calgary, AB T2N 1N4, Canada; hyunjae.lee@ucalgary.ca (H.J.L.); justin.rejimone@ucalgary.ca (J.R.);; 2Department of Biomedical Engineering, Schulich School of Engineering, University of Calgary, Calgary, AB T2N 1N4, Canada

**Keywords:** carbon nanotube, silicone elastomer, conductive ink, high dispersibility, piezoresistive sensing

## Abstract

An alternative conductive ink based on carbon nanotubes (CNTs) was developed using a platinum-catalyzed silicone elastomer and isopropyl alcohol (IPA). The inclusion of IPA in the conductive CNT ink facilitated the optimization of its mechanical strength, electrical conductivity, and viscosity. Compared to conventional silicone rubber-based conductive polymers that often solidify in a few hours at room temperature or with heating, this liquid composite of CNT particles and IPA exhibited a prolonged duration of up to several months in a hermetic environment, maintaining chemical stability even with the elastomer and its curing agent. The gradual evaporation of IPA initiates a well-known cross-linking process, leading to stretchability and electrical conductivity derived from the silicone elastomer and CNT particles, respectively. The relationship between the mechanical strength and electrical conductivity of the hardened conductive CNT ink was studied, which helped determine the optimized concentration of CNT particles in the conductive CNT ink. Subsequently, a piezoresistive sensor was designed, fabricated, and evaluated. The conductive CNT ink-based piezoresistive sensor showed high sensitivity and good repeatability with respect to a wide range of external forces. The effect of the concentration of CNT particles on the viscosity of the conductive CNT ink was also investigated, providing a better understanding of the entanglement of CNT particles within the silicone elastomer. A coating test using the conductive CNT ink with a paper cutting machine demonstrated its potential for adaptation to various printing techniques, including screen printing. The proposed conductive CNT ink, characterized by a simple chemical composition, facile fabrication process, use of non-toxic elements, high electrical conductivity, and stretchability, combined with an extended duration, has the potential to be applied for multiple purposes, such as various types of flexible and wearable electronics.

## 1. Introduction

Flexible and wearable electronics have garnered significant interest for numerous applications, including human activity and health monitoring, soft and biomimetic robots, and various sensors for environmental perception [1,2,3,4,5]. The success of these applications requires reliable, refined, and precise fabrication techniques, as well as stable electrical and mechanical properties of conductive elements. Carbon nanotubes (CNTs), in particular, are widely utilized as a conductive material owing to their outstanding electrical and mechanical properties [6,7,8]. To effectively deposit CNT particles on specific areas, CNTs are often prepared in the form of conductive ink. For instance, CNTs can be mixed with various solvents to achieve homogeneous dispersion of particles on thin films or targeted areas [9,10]. However, the dispersibility of CNTs is limited in solvents, often restricting the concentration of CNTs in the solution because excessive amounts of CNTs cannot be easily dispersed. Additionally, a higher concentration of CNTs in IPA can degrade the quality of conductive films, leading to cracks after the coating process [10]. To prevent these problems, various silicone elastomers are used to fabricate conductive and flexible thin films [11,12]. In this case, achieving a balance between high electrical conductivity and reliable mechanical properties is challenging because a higher concentration of CNTs increases the viscosity of the uncured silicone elastomer–CNT mixture, hindering the proper deposition of CNT particles on film surfaces after curing [12]. Moreover, once these silicone elastomers are mixed with curing agents, the cross-linking process begins immediately, which can lead to a non-homogeneous dispersion of CNT particles within the curing silicone elastomers. In other words, long-term storage of the silicone elastomers and curing agents as a single composite is not feasible, implying that the preliminary mixing of these two materials is required each time before use. Therefore, prolonging the cross-linking process of the silicone elastomers would be advantageous for various purposes. One solution would be the addition of a material that acts as a temporary inhibitor of the irreversible chemical reaction between a silicone elastomer and a curing agent, allowing the composite of these two materials to remain liquid for a longer time than in conventional use.

Similar to silicone elastomer-based conductive polymers, various types of conductive inks have also been reported, where key factors were analyzed, such as electrical conductivity, adhesion, flexibility, stability, viscosity, cost, and safety. Several studies have explored the development of high-performance conductive inks using CNTs [13,14,15]. For instance, the cresol-based CNT solution exhibits polymer-like processability with tunable rheological and viscoelastic properties, as well as chemical stability [13]. However, cresols, which are considered toxic, require removal through heating or washing, which introduces an additional step to the fabrication process. CNT ink reported in another study primarily involves hydroxypropyl methyl cellulose (HPMC) and γ-aminopropyl triethoxy silane (APTES) [14]. But this ink requires complex pretreatments, which could be a drawback. A composite based on graphene, CNTs, and a silicone elastomer was also developed for direct ink writing (DIW) [15]. Although this composite demonstrates excellent electrical stability with a near-zero temperature coefficient of resistance (TCR), the mixture of the silicone elastomer and its curing agent offers a limited processing window for extended printing due to a rapid cross-linking process.

In this study, we present an alternative conductive CNT ink with a significantly extended shelf life. To enhance electrical conductivity and achieve good dispersibility, multi-walled CNT (MWCNT) powder was mixed with IPA and Ecoflex, which is a platinum-catalyzed silicone elastomer. Various concentrations of MWCNTs in the CNT ink were tested to analyze chemical, electrical, and mechanical properties. We discovered that IPA regulates the viscosity of the CNT solution, temporarily inhibits the cross-linking process of uncured Ecoflex, and facilitates adequate absorption of CNT particles. The volatility of IPA facilitates rapid solidification of the conductive CNT ink at room temperature. Additionally, we evaluated the applicability of the CNT ink for piezoresistive sensing by fabricating a sensor that exhibited good reproducibility and high sensitivity. We believe that the developed conductive CNT ink, containing IPA and Ecoflex, offers a compelling alternative to conventional conductive polymers and inks due to a simple fabrication process, minimum components, excellent electrical and mechanical properties, and long-term stable storage in a liquid state, making it suitable for various applications in flexible and wearable electronics.

## 2. Materials and Methods

### 2.1. Materials

Multi-walled carbon nanotubes (MWCNTs, >99 wt%, 13–18 nm outer diameter, 3–30 μm length) were purchased from Cheap Tubes (Grafton, VT, USA), and Ecoflex 00-30 was obtained from Smooth-On (Macungie, PA, USA). Isopropyl alcohol (IPA) with 99.5% purity was purchased from VWR International (Radnor, PA, USA).

### 2.2. Preparation of the Conductive CNT Ink

IPA (5 mL) and MWCNTs (0.3 g) were manually mixed and stirred for 10 min in a Petri dish at room temperature. Simultaneously, 5 mL of Ecoflex 00-30, prepared by mixing parts A and B in a 1:1 ratio, was placed in a conical tube. The uncured Ecoflex was then poured into the mixture of IPA and MWCNTs and subsequently mixed mechanically using a blender for 10 min at room temperature to avoid the unexpectedly accelerated solidification of Ecoflex caused by heat-releasing methods such as ultrasonication.

### 2.3. Optical Imaging of the Conductive CNT Ink-Coated Surfaces

A mold with dimensions of 27 mm × 18.6 mm × 5 mm for sample preparation was designed using AutoCAD 2023 and fabricated using a 3D printer (X-Plus, Qidi Tech, Ruian, China). Subsequently, 1 mL of Ecoflex 00-30 was poured into the mold after manual stirring for 10 min at room temperature. The uncured Ecoflex in the mold was stored and cured at 70 °C for 3 h in a vacuum oven (Cole-Parmer, Vernon Hills, IL, USA). Following curing, 60 μL of the conductive CNT ink with varying chemical compositions (CNT concentrations of 15.7, 17.7, 24.4, 27.2, and 30.1 wt%) was dispensed onto each sample structure made with the cured Ecoflex and dried for 4 h at room temperature. The area coated with the conductive CNT ink was designed to have dimensions of 6 mm × 6 mm × 3 mm.

### 2.4. Fourier-Transform Infrared (FTIR) Spectroscopy

Fresh samples with different compositions were prepared (IPA only, Ecoflex only, IPA-Ecoflex composite, IPA-Ecoflex-MWCNT composite) as liquid mixtures to avoid irregularities caused by the evaporation of IPA and the solidification of Ecoflex. For all of the samples, 5 mL IPA, 5 mL Ecoflex, and 3 g MWCNT were selectively used. Approximately 300 μL of each prepared sample was loaded onto the metal holder of the FTIR spectrometer (Thermo Electron Corporation, Waltham, MA, USA) for analysis.

### 2.5. Tensile Strength Measurement

A mold with dimensions of 27 mm × 18.6 mm × 5 mm was designed using AutoCAD 2023 and fabricated using the aforementioned 3D printer. Then, 1 mL of the CNT ink with various combinations was poured into the mold and dried for 4 h at room temperature. For tensile strength measurement, a mechanical tester (ESM303, Mark-10, Copiague, NY, USA) was used. Each sample, with dimensions of 23 mm × 14.6 mm × 3 mm, was stretched at a rate of 10 mm/min until a displacement of 20 mm was reached.

### 2.6. Electrical Resistance Measurement

Each 60 μL, containing varying quantities of MWCNT particles and either with or without IPA (concentrations of 1.8, 3.6, 2.1, 3.1, and 4.1 wt%), was dispensed into the aforementioned mold. After drying for 4 h at room temperature, the surface was covered with uncured Ecoflex, and the samples were left for an additional 4 h at room temperature. Subsequently, the samples were connected to a potentiostat (PalmSens4, Houten, The Netherlands), and their electrical resistance was recorded at 1-s intervals for 60 s for each sample.

### 2.7. Compressive Force and Electrical Resistance

The samples prepared for electrical resistance measurement were used to analyze the relationship between compressive force and its effect on the electrical resistance of the samples as piezoresistive sensors. A force sensor from the mechanical tester was moved at 20 mm/min for three cycles, with force limits of 0.049, 0.098, and 0.200 N, respectively. Concurrently, electrical resistance changes in the piezoresistive sensor samples caused by the applied compressive force were recorded with the potentiostat at 1-s intervals.

### 2.8. Analysis of Rheological Properties

A digital viscometer (NDJ-5S, Ato, Fukuoka, Japan) was used to analyze the rheological properties of the three distinct conductive CNT ink samples containing 0.3 g, 0.4 g, and 0.5 g of MWCNTs, respectively, in 5 mL of IPA and 5 mL of Ecoflex. The viscosity measurement was conducted using the digital viscometer at 60 rpm for 1 min per sample.

## 3. Results and Discussion

### 3.1. Optical Images of CNT Ink Samples with Various Concentrations of MWCNTs

When Ecoflex and its curing agent are mixed with MWCNTs in the presence of IPA, their cross-linking process is not initiated immediately, primarily due to the inhibitory effect of IPA between these two components. Although the exact mechanism by which IPA hinders the cross-linking process of Ecoflex remains unclear, it was observed that the conductive CNT ink stored in an airtight condition can remain liquid for several months, with IPA acting as a temporary inhibitor of the cross-linking process. A similar result was reported with the combination of carbon black, IPA, and Ecoflex [16]. Upon exposure to air, the conductive CNT ink begins to harden due to the evaporation of IPA and the activation of the cross-linking process of Ecoflex (Figure 1a). Previously, it has been reported that the vinyl group of siloxane oligomers in Ecoflex reacts with a cross-linker in the curing agent to impart elasticity to the solidified Ecoflex [17,18]. Figure 1b,c show a fabricated conductive CNT ink sample and a piezoresistive sensor based on the conductive CNT ink, respectively. The initial consistency of the conductive CNT ink is similar to that of clay, which is transformed into an elastic texture after solidification. The role of each component in the conductive CNT ink is straightforward to understand: IPA for the temporary inhibition of the cross-linking process of the silicone elastomer, CNT particles for enhanced electrical properties, and Ecoflex for improved mechanical properties.

Another function of IPA is to adjust the viscosity of the conductive CNT ink, which is also related to the quantity of MWCNTs present. MWCNTs are well known for their high surface area, facilitating interaction with uncured Ecoflex. If the concentration of MWCNTs in the conductive CNT ink increases, the entanglement of MWCNTs within the uncured Ecoflex matrix is reinforced, which is believed to reduce the fluidity of the conductive CNT ink and increase its viscosity. Using 5 mL of IPA and 5 mL of uncured Ecoflex (including its curing agent), different amounts of MWCNTs, ranging from 0.1 g to 0.4 g, were tested to clarify the visual difference between each sample (Figure 2a). When no IPA was added to the samples containing 0.1 g and 0.2 g MWCNTs (concentrations of approximately 1.8 and 3.6 wt%, respectively), their surfaces were smooth due to the absence of IPA displacement during the cross-linking process. In contrast, the samples containing 0.2 g, 0.3 g, and 0.4 g MWCNTs with IPA (concentrations of approximately 2.1, 3.1, and 4.1 wt%, respectively) showed bumpy surfaces attributed to the evaporation of IPA during gradual solidification. If the concentration of MWCNTs is not sufficiently high, the hardened conductive CNT ink will not exhibit high electrical conductivity. However, an excessively high concentration of MWCNTs would render the fluid highly viscous, eventually leading to the transformation of the conductive CNT ink into a powder-like state. Therefore, the addition of IPA is a useful method to achieve an adjusted fluidic state of the conductive CNT ink and to ensure reliable electrical conductivity. The relationship between the addition of IPA and electrical conductivity was explained in the electrical resistance measurement . Figure 2b,c are a scanning electron microscopic (SEM) image of the conductive CNT ink with 0.3 g MWCNTs and its elemental analysis based on energy dispersive spectroscopy (EDS). Compared to O and C, Si showed the highest peak, which is possibly related to a technical factor of the equipment in combination with its atomic weight. Another thing to note is that given that not every MWCNT particle was on the surface, the analysis was limited because Ecoflex is not conductive. Nevertheless, the results indicated that the main elements of the conductive CNT on the surface layers were carbon, oxygen, and silicon, as expected.

### 3.2. Chemical Analysis of the CNT Ink Elements and Their Chemical Structures

An FTIR analysis was conducted to identify the components of the conductive CNT ink, as shown in Figure 3. Four different samples were examined to elucidate probable chemical reactions between IPA, Ecoflex, and MWCNTs. Figure 3a–c revealed that the spectra of each bond were maintained with slight modifications in light absorbance after the mixture of IPA and Ecoflex. In other words, the O-H band from IPA located between 3000 and 4000 cm^−1^, the Si-O-SI and Si-CH_3_ bands from Ecoflex located between 1000 and 1100 cm^−1^, and in 2960 cm^−1^, respectively, remained in the mixture of IPA and Ecoflex. Regarding modifications in light absorbance, it is assumed that several factors such as concentration changes and peak overlaps could affect this result. When MWCNTs were added to this mixture, the light absorbance of certain existing spectra decreased or increased without the appearance of additional spectra. This implies that the conductive CNT ink was likely formed through the physical mixture of each component rather than chemical reactions that would create new bonds. In other words, the constituent elements of the conductive CNT ink retain their chemical stability and act independently, allowing for the characteristic expression of each element forming electrical, chemical, and mechanical properties.

### 3.3. Tensile Strength Measurement with Various Concentrations of MWCNTs

To understand the relationship between tensile strength and the concentration of MWCNTs in the conductive CNT ink, with or without IPA, five distinct samples were prepared using a 3D-printed mold, and their tensile strengths were measured using a mechanical tester (Figure 4a,b). The mechanical properties of the solidified conductive CNT ink were found to be related to the concentration of MWCNTs and that of IPA (Figure 4c). As the concentration of MWCNTs increases in the dried conductive CNT ink, the stress resisted with the same strain decreases, whereas the viscosity increases in the conductive CNT ink. It is assumed that the enhanced entanglement of MWCNTs within the elastic network of uncured Ecoflex weakens the physical bonding of the elements of the conductive CNT ink, resulting in lower stress at the same strain. Similarly, as strain increases, samples with higher concentrations of MWCNTs exhibit lower endurable stress. However, the sample containing 0.4 g MWCNTs with IPA (4.1 wt%) did not show a consistent result, likely because the excessively high concentration of MWCNTs in the conductive CNT ink hindered the proper dispersion of CNT particles, leading to irregular mechanical properties with increasing strain. The presence of IPA also affected the relationship between strain and stress in the dried conductive CNT ink samples. For example, the ink sample containing 0.2 g MWCNTs with IPA (2.1 wt%) had lower viscosity than the sample without IPA (3.6 wt%), which resulted in reduced stress at the same strain. This result suggests that the increased volume of IPA may weaken the physical bonding of the elements of the conductive CNT ink in a different manner compared to the increased concentration of MWCNTs.

### 3.4. Electrical Resistance Measurement in the Form of a Piezoresistive Sensor

A piezoresistive sensor was fabricated using Ecoflex to create an elastic outer structure. The surface of the solidified conductive CNT ink was then covered with the same silicone elastomer after being poured onto the outer structure (Figure 5a(i,ii). It is assumed that electrical conductivity is achieved through a percolating network formed by MWCNTs and Ecoflex. Dried at room temperature, the conductive CNT ink was surrounded by a protective layer to ensure its reliable and reproducible functioning as a piezoresistive sensor. Five different samples were prepared with varied concentrations of MWCNTs, both with and without IPA (Figure 5b(i)). For all samples containing IPA, the same amount of IPA was used to clarify its role in the conductive CNT ink with different amounts of MWCNTs. After measurement, the results revealed electric resistance values of 9.2 MΩ for the sample containing 0.1 g MWCNTs without IPA (1.8 wt%) and 8.1 MΩ for the sample containing 0.2 g MWCNTs without IPA (3.6 wt%), respectively (Figure 5b(ii)). The sample with 0.1 g MWCNTs was sufficiently fluidic without IPA, indicating that the addition of IPA in this case could excessively dilute the entire composite, increase the distance between MWCNT particles, and hinder proper solidification. That was the reason why no IPA was added to the sample with 0.1 g MWCNTs. The second sample exhibited high irregularities in electric resistance, which might have been caused by the uneven distribution of CNT particles due to its increased viscosity compared to the first sample, which was similar to a powder-like state. The third sample, containing 0.2 g MWCNTs with IPA (2.1 wt%), had lower electrical resistance with reduced irregularities, demonstrating the necessity of IPA for highly viscous ink samples. As the concentration of MWCNTs increased from 0.3 g to 0.4 g with IPA (3.1, and 4.1 wt%, respectively), the average electrical resistances were measured as 200 kΩ and 25 kΩ, respectively. This result suggests that higher amounts of MWCNTs facilitate effective electron transfer within the non-conductive elastic network of the dried conductive CNT ink.

**Figure 5 micromachines-17-00014-f005:**
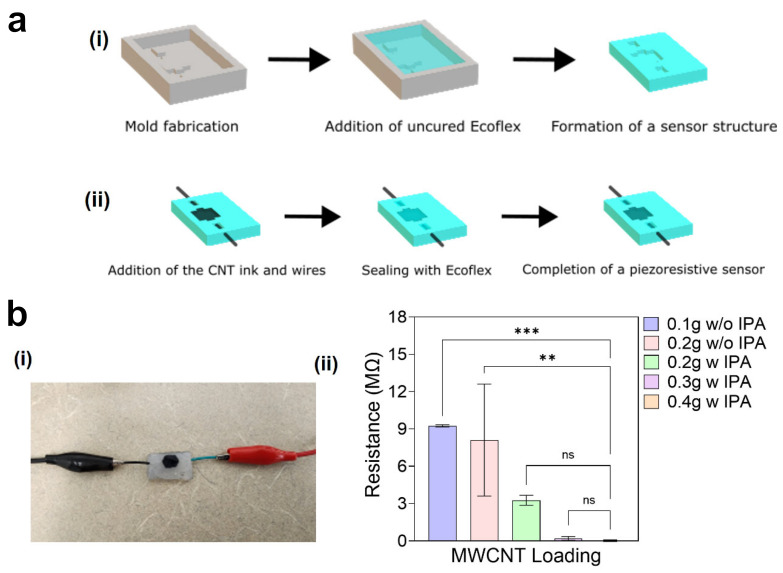
Preparation of a piezoresistive sensor sample. (**a**) Fabrication process: (**i**) elastic structure, (**ii**) entire piezoresistive sensor. (**b**) Electrical analysis: (**i**) fabricated sample, (**ii**) electrical resistance measurement. ***: *p* < 0.001, **: *p* < 0.01, and ns: no signicant.

### 3.5. Compressive Force Measurement as a Piezoresistive Sensor

The sample containing 0.3 g MWCNTs with IPA (3.1 wt%) was selected to conduct compressive force measurement as a piezoresistive sensor because it exhibited high electrical conductivity and stable mechanical properties with respect to elongation. When the set force was 0.049 N, the electrical resistance of the sample showed an approximately 13% increase with an approximate displacement of 0.13 mm (Figure 6a(i,ii)), whereas the measured compressive force averaged 0.072 N. With a set force of 0.098 N, approximately a 24% increase in electrical resistance was recorded from the sample, with an approximate displacement of 0.19 mm (Figure 6b(i,ii)). At that time, the measured compressive force was approximately 0.124 N. A set force of 0.200 N, converted to an average measured force of 0.234 N, showed that an approximate displacement of 0.29 mm generated approximately 38% increase in electrical resistance (Figure 6c(i,ii). The difference between the set forces and the measured forces was due to a technical issue between the software and the equipment. Based on the results of compressive force measurement, the solidified conductive CNT ink for a piezoresistive sensor can detect approximately 0.050 N as the minimum force. However, if the external force increases exponentially, the CNT particles encapsulated in the piezoresistive sensor may be significantly disturbed, possibly due to a small portion of IPA remaining inside, which may affect its stability and repeatability. Therefore, the conductive CNT ink-based piezoresistive sensor is better suited for detecting small forces, specifically those less than 0.200 N. This result also implies that CNTs without functionalization can be utilized for piezoresistive sensing with additional elaborations.

### 3.6. Investigation of Rheological Properties and a Coating Test

The optimization of the rheological properties of the conductive CNT ink is indispensable for the effective fabrication of thin films. Conductive CNT inks with both excessively high and low viscosities can hinder their proper fixation on a targeted area. For this reason, a rheological analysis of the conductive CNT ink was conducted with three distinct samples consisting of 0.2, 0.3, and 0.4 g MWCNTs, with concentrations of 2.1, 3.1, and 4.1 wt%, respectively, along with 5 mL of Ecoflex and IPA (Figure 7a). In the range of 0.2 to 0.4 g MWCNTs, the viscosity of the samples increased from 629 to 801 mPa·s (Figure 7b). In conclusion, the quantity of MWCNTs in the conductive CNT ink was found to be proportional to its viscosity, providing useful information for optimizing its rheological properties.

After the rheological analysis, a coating test using the conductive CNT ink was also conducted with various substrates, including cured Ecoflex, a sheet of paper, and a Petri dish, as one of the potential applications. The objective of this test was to evaluate the cohesiveness of the conductive CNT ink after its solidification onto the substrate. A machine called Cricut was used to create an interdigitated pattern using a paper mold (Figure 8a,b). After pouring the conductive CNT ink onto the paper mold, the mold was removed from the substrate (Figure 8c,d). The width of the interdigitated pattern was approximately 2 mm. Among the distinct substrates, the patterned conductive CNT ink showed the best performance on the cured Ecoflex. The sheet of paper was not suitable for the conductive CNT ink-based coating due to its liquid-absorbing properties. The coating with the Petri dish was not well realized due to the lack of adhesiveness with the conductive CNT ink. This result indicates that the conductive CNT ink is most compatible with a cured Ecoflex substrate because one of the constituent elements is the same silicone elastomer. This information would be useful for the future application of the conductive CNT ink in various patterned piezoresistive sensors based on silicone elastomers and conductive materials. It can be concluded that the conductive CNT ink has potential in several printing techniques, such as stencil printing and blade coating. Further characterizations in terms of printing accuracy and electrical resistivity could be elaborated in the continuous study.

## 4. Conclusions

The optimal concentration of the conductive CNT ink was studied considering the presence of MWCNTs, Ecoflex, and IPA. The optical characteristics of the dried conductive CNT ink surface were also analyzed, revealing that an increased volume of IPA enhanced the porosity of the solidified ink, a characteristic that could be reduced by increasing the amount of MWCNTs. However, the porosity of the solidified ink could also be affected by several factors, such as dispersion, aggregation, and the way it solidifies. According to the FTIR analysis, Ecoflex, its curing agent, IPA, and MWCNTs were likely physically mixed without chemical reorganization, implying that each element can independently contribute distinct material characteristics to the conductive CNT ink. Tensile strength measurements indicated that the composites containing 0.2 and 0.3 g MWCNTs with IPA and Ecoflex (2.1 and 3.1 wt%, respectively) exhibited stable mechanical properties under vertically directional deformation. Electrical resistance measurements of the conductive CNT ink-based piezoresistive sensors elucidated that the composites with 0.3 and 0.4 g MWCNTs, including IPA (3.1 and 4.1 wt%, respectively), possessed excellent electrical conductivity ranging from a few tens to a few hundreds of kilo-ohms. Compressive force measurements clarified that the conductive CNT ink-based piezoresistive sensor demonstrated high sensitivity and reliability, especially when the external force was less than 0.15 N. The viscosity of the conductive CNT ink was proportional to the amount of MWCNTs, ranging approximately from 630 to 800 mPa·s between 0.2 and 0.4 g MWCNTs. The adhesiveness of the conductive CNT ink was also confirmed using a paper-based pattern created using a cutting machine, showing the strongest adhesion to the substrate made with cured Ecoflex. The applicability of this newly proposed conductive CNT ink, characterized by a simple fabrication process, a minimized material composition, and an extended liquid state duration, holds promise for numerous applications, mainly including flexible and wearable electronics such as robotic finger-related components.

## Figures and Tables

**Figure 1 micromachines-17-00014-f001:**
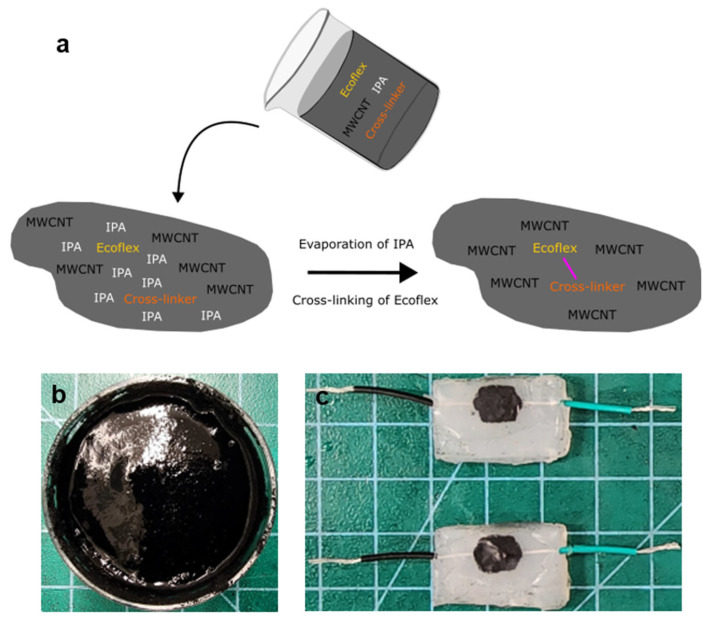
Conceptual image of the conductive CNT ink. (**a**) Formation of the conductive CNT ink. (**b**) Prepared conductive CNT ink sample. (**c**) Piezoresistive sensor sample combined with an elastic material.

**Figure 2 micromachines-17-00014-f002:**
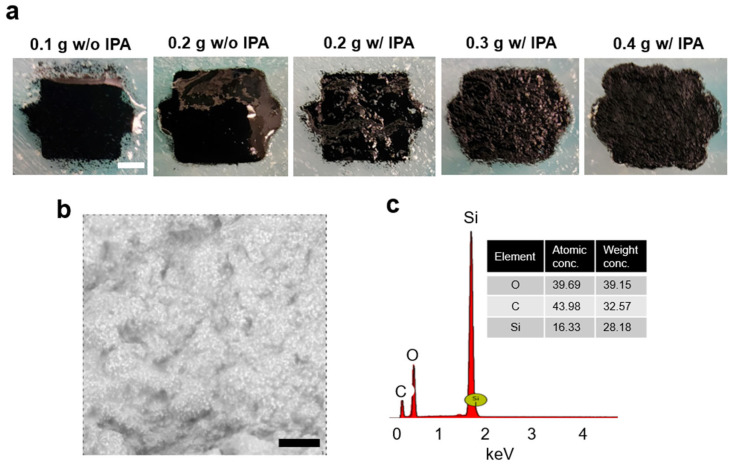
Visualization of the conductive CNT ink. (**a**) Optical images of MWCNT 0.1 g without IPA (1.8 wt%), MWCNT 0.2 g without IPA (3.6 wt%), MWCNT 0.2 g with IPA (2.1 wt%), MWCNT 0.3 g with IPA (3.1 wt%), and MWCNT 0.4 g with IPA (4.1 wt%). Scale bar = 2 mm. (**b**) SEM image of a hardened sample of MWCNT 0.3 g with IPA. Scale bar = 100 μm. (**c**) Element analysis result.

**Figure 3 micromachines-17-00014-f003:**
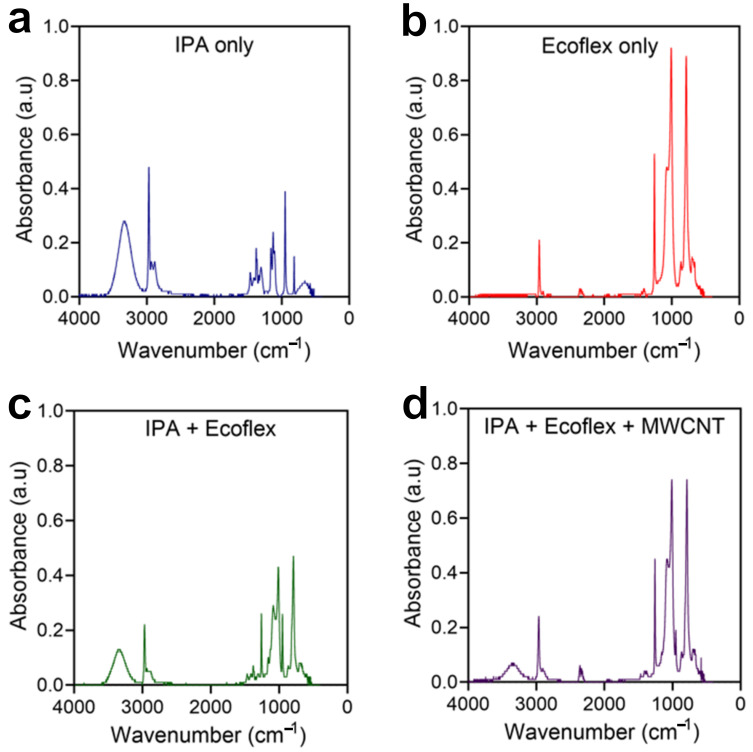
FTIR analysis. (**a**) IPA only. (**b**) Ecoflex only. (**c**) IPA + Ecoflex. (**d**) IPA + Ecoflex + MWCNTs.

**Figure 4 micromachines-17-00014-f004:**
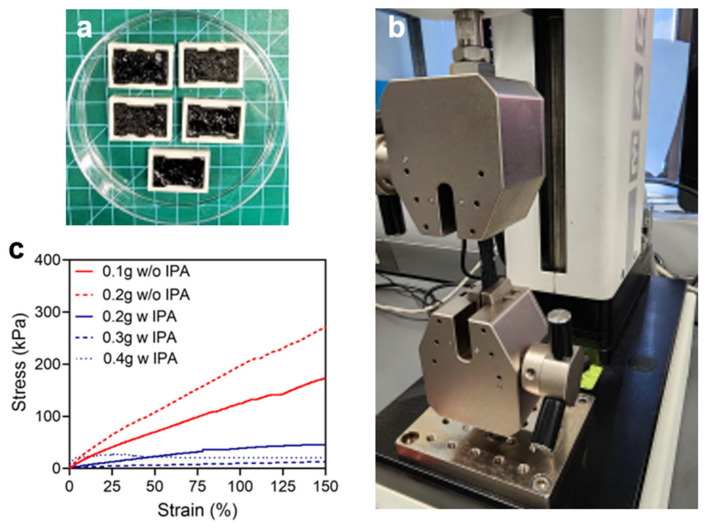
Mechanical analysis. (**a**) Mechanical tester. (**b**) Solidified CNT ink samples. (**c**) Relationship between strain and stress.

**Figure 6 micromachines-17-00014-f006:**
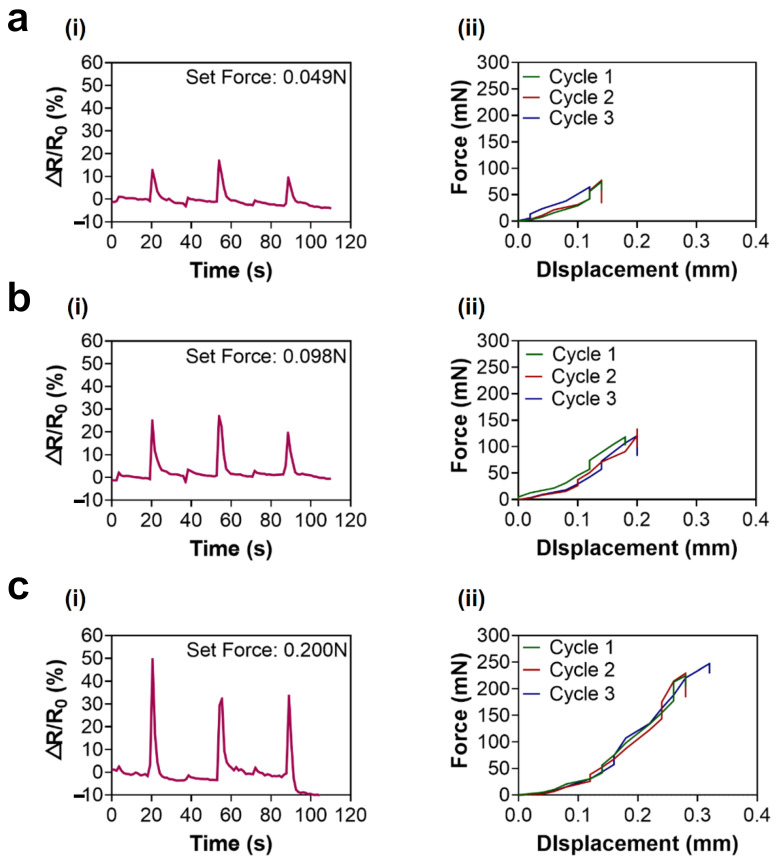
Compressive force measurement using samples made with MWCNT 0.3 g with IPA when the set force was (**a**) 0.049 N. Relationship between (**i**) time and resistance, (**ii**) displacement and force. (**b**) 0.098 N. Relationship between (**i**) time and resistance, (**ii**) displacement and force. (**c**) 0.200 N. Relationship between (**i**) time and resistance, (**ii**) displacement and force.

**Figure 7 micromachines-17-00014-f007:**
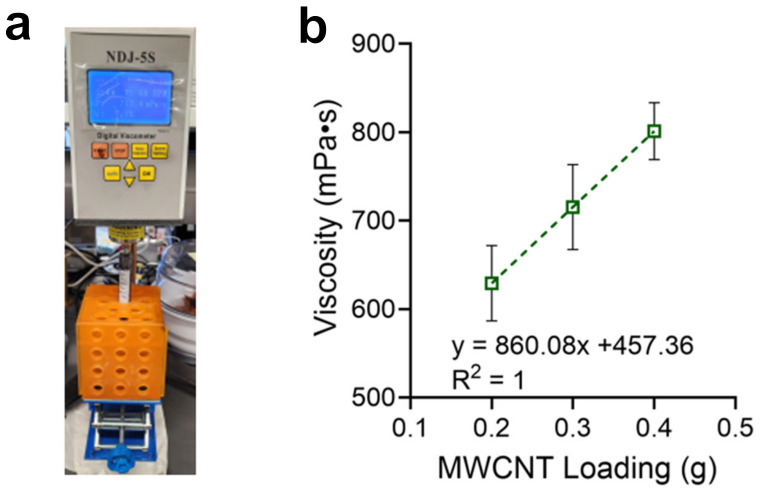
Rheological analysis. (**a**) Viscometer with a sample. (**b**) Relationship between the concentration of MWCNTs in the conductive CNT ink samples and their viscosities.

**Figure 8 micromachines-17-00014-f008:**
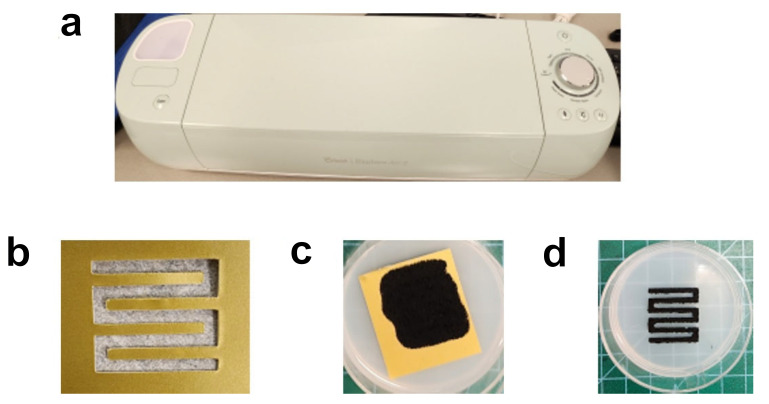
CNT ink coating. (**a**) Machine for the formation of a paper mold. (**b**) Paper mold covered with the CNT ink. (**c**) Patterned paper mold. (**d**) CNT ink-based printed pattern.

## Data Availability

The original contributions presented in this study are included in the article. Further inquiries can be directed to the corresponding author.

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
