# Peer review of "Highly Dispersible and Stable Carbon Nanotube Ink with Silicone Elastomer for Piezoresistive Sensing"

_micromachines, 2025, doi:10.3390/mi17010014_

Round 1

Reviewer 1 Report

Comments and Suggestions for Authors

In this article, Highly Dispersible and Stable Carbon NanotubeInk with Sil-icone Elastomer for Piezoresistive Sensing. The authors designed a CNT based ink. They have performed thorough characterization of the ink including FTIR,  tensile strength, electrical resistance and also measured piezoelectric sensing properties. The article is well structured and very well written with clear scientific rigor.

I would recommend it's publication with one small revision. 

1. If possible please include a table with previous studies related to CNT ink used for DIW along with their key properties and applications. 

Author Response

Reviewer #1: In this article, Highly Dispersible and Stable Carbon NanotubeInk with Sil-icone Elastomer for Piezoresistive Sensing. The authors designed a CNT based ink. They have performed thorough characterization of the ink including FTIR,  tensile strength, electrical resistance and also measured piezoelectric sensing properties. The article is well structured and very well written with clear scientific rigor.

I would recommend it's publication with one small revision. 

1) If possible please include a table with previous studies related to CNT ink used for DIW along with their key properties and applications. 

Response: The authors appreciate the reviewer’s suggestion. In this study, the authors intended to emphasize the simplicity in terms of fabrication process and chemical composition in combination with high electrical conductivity, good stretchability, and long-term storage in the form of liquid. For this reason, the authors took the references #13-15 about the conductive inks that were similar to these conditions using CNTs [page #14]. However, except the reference #15, the two references were better adapted to different printing methods such as stencil printing and blade coating. In addition, at the end of the manuscript, the authors included an example of stencil printing using a machine called Cricut. Therefore, the authors think that it would not be suitable to put a table with previous studies related to CNT inks used for direct ink writing. The authors added this discussion in the revised manuscript [page #12] as follows,

 It can be concluded that the conductive CNT ink has potential in several printing techniques such as stencil printing and blade coating.

Reviewer 2 Report

Comments and Suggestions for Authors

This work proposes a method for significantly extending the shelf life of carbon nanotube (CNT)/silicone elastomer (Ecoflex) composite inks using isopropanol (IPA) and demonstrates its application in piezoresistive sensing. While this study has a good starting point and the preliminary results show potential, several areas require further clarification and exploration:
1. In the pressure test shown in Figure 6, there is a significant difference between the “Set Force” and the “Measured Force” (e.g., a set force of 0.049 N and a measured average of 0.072 N). The authors should explain the cause of this difference.
2. The coating test in Figure 8 is inadequate. As a paper on “ink,” in addition to showing the pattern, key printing/coating performance parameters, such as the accuracy of the printed lines and resistivity, should be provided.
3. The core claim of "long-term stability" (up to several months) lacks experimental data support. All test data appear to come from freshly prepared inks. The authors should supplement related data.

Author Response

Reviewer #2: This work proposes a method for significantly extending the shelf life of carbon nanotube (CNT)/silicone elastomer (Ecoflex) composite inks using isopropanol (IPA) and demonstrates its application in piezoresistive sensing. While this study has a good starting point and the preliminary results show potential, several areas require further clarification and exploration:

1) In the pressure test shown in Figure 6, there is a significant difference between the “Set Force” and the “Measured Force” (e.g., a set force of 0.049 N and a measured average of 0.072 N). The authors should explain the cause of this difference.

Response:  The authors appreciate the reviewer’s comments. The authors think that the difference observed between the set force and the measured force is due to a technical issue between the software and the equipment. However, the authors decided to share the detailed information for a better understanding from the future reader. The authors added this explanation in the revised manuscript [page #10] as follows,

 The difference between the set forces and the measured forces was due to a technical issue between the soft and the equipment.

 2) The coating test in Figure 8 is inadequate. As a paper on “ink,” in addition to showing the pattern, key printing/coating performance parameters, such as the accuracy of the printed lines and resistivity, should be provided.

Response: The authors appreciate the reviewer’s comment. In this study, the authors intended to show the conductive CNT ink with the simplest chemical composition. To understand the characteristics of the newly suggest conductive ink, the authors conducted chemical, mechanical, and electrical analyses. For the coating test, the authors took an example of the references #13-14 where stencil printing samples were shown. The main goal of this coating test was to verify the compatibility of distinct substrates such as paper sheet, petri dish, and cured Ecoflex. The detailed characterization of the printed pattern including accuracy and electrical resistivity could be analyzed in the continuous study. The authors added this discussion in the revised manuscript [page #12] as follows,

 Further characterizations in terms of printing accuracy and electrical resistivity could be elaborated in the continuous study.

 3)  The core claim of "long-term stability" (up to several months) lacks experimental data support. All test data appear to come from freshly prepared inks. The authors should supplement related data.

Response: The authors appreciate the reviewer’s comment. For the chemical, mechanical, and electrical analyses, the authors utilized freshly prepared inks. As an additional feature, the authors found that the conductive CNT ink is also capable to remain liquid for several months. In addition, a similar study was reported with the combination of carbon black, IPA, and Ecoflex [page #13]. The authors mentioned this information in the revised manuscript [page #5, 1st paragraph]. It is believed that for the additional analysis regarding the material properties of the long-time stored conductive CNT ink could be treated in the continuous study as follows,

 A similar result was reported with the combination of carbon black, IPA, and Ecoflex [16].

Reviewer 3 Report

Comments and Suggestions for Authors

The article titled “Highly Dispersible and Stable Carbon Nanotube Ink with Silicone Elastomer for Piezoresistive Sensing” by Lee et al. discuss development of conductive ink based on carbon nanotubes (CNTs), platinum-catalyzed silicone elastomer and isopropyl alcohol (IPA). The work presents a potentially high-performing material, and authors claims are exceptional. However, there are few section which may be improved for making this work more comprehensive. Below are the specific points that need attention:

  1. Pg2 Line 44: “For instance, CNTs can be mixed with various solvents to achieve homogeneous dispersion of particles on thin films or targeted areas [9,10]. However, the dispersibility of CNTs is limited in certain solvents, often restricting the concentration of CNTs in the solution.” These two statements are contradictory. Kindly improve this.
  2. Pg2 Line 64: “Similar to silicone elastomer-based conductive polymers, several types of conductive inks have also been reported, where key factors include, with key attributes being electrical conductivity, adhesion, flexibility, stability, viscosity, cost, and safety.” The language may be improved here.
  3. Pg3 section 2.2. Preparation of the conductive CNT ink: in accordance with literature MWCNT remains in bundle form and requires ultrasonication to disperse with any silicon based polymer. Authors should explain how simple manual mixing and stirring formed uniform dispersion of CNT and Ecoflex?
  4. Authors should explain, if mixing of CNT is simple, why most literature suggest functionalization, or oxidation of CNTs before making its dispersion with any silicone based polymer?
  5. Pg5 Line 178: “When no IPA was added to the samples containing 0.1 g and 0.2 g MWCNTs (concentrations of approximately 15.7 and 27.2 wt%, respectively)”. Kinldy discuss how wt% is derived here, is it w.r.t. IPA or Ecoflex or combined?
  6. Pg6 Line 185: “It is worth noting that if the concentration of MWCNTs is not sufficiently high, the hardened conductive CNT ink will not exhibit high electrical conductivity. However, an excessively high concentration of MWCNTs would render the fluid highly viscous, eventually leading to the transformation of the conductive CNT ink into a powder-like state. Therefore, the addition of IPA is a useful method to achieve an adjusted fluidic state of the conductive CNT ink and to ensure reliable electrical conduc-tivity.” Kindly provide concentration dependent electrical conductivity data to support this claim.
  7. Pg6 Figure 2b and c, do not provide any significant information. SEM image is not clear to distinguish between CNT and Ecoflex. The elemental analysis graph is showing largest peak for Si, still it has a wt conc. of 28.18.

Kindly explain this and use another SEM distinguishing CNT and polymer. It cannot tell you if the CNTs are well-dispersed, aggregated, or oriented in a specific way. This is critical for electrical and mechanical properties. A poorly dispersed sample would have terrible properties.

  1. Pg 7 Figure 3: Authors should add FTIR of MWCNT in comparison with other 4 spectra. Authors should discuss in detail about specific peaks from the Ecoflex (like Si-O-Si, Si-CH3) or IPA (O-H) monitored for significant shifts or shape changes that would indicate even weak interactions like hydrogen bonding between SWCNT and polymer?
  2. Authors have identified modifications in light absorbance (Line 204). Kindly quantify these changes (like, peak shifts in wavenumbers, percentage change in intensity) and discuss their potential physical origin, such as hydrogen bonding or scattering effects from the MWCNT.
  3. Authors should discuss how IPA + Ecoflex sample for FTIR prepared. Since IPA is a volatile solvent and Ecoflex cures, was this spectrum taken from a liquid mixture, a dried film, or a cured elastomer? Please clarify the sample state for all spectra, as this have a major affect on the interpretation.
  4. Mechanical properties of CNT Ecoflex depend on the dispersion and interface between the CNTs and the Ecoflex. Authors should provide data microscopy (SEM/TEM) or Raman mapping to corroborate the physical state of dispersion and quantify the bundle size of CNT.
  5. Authors should propose the mechanism for the electrical and mechanical properties in CNT Ecoflex. Like, is the electrical conductivity achieved through a percolating network of CNTs within the Ecoflex matrix, and how does the IPA facilitate its formation before evaporating?
  6. Authors should demonstrate how morphology of CNTs changes with multiple stretching cycles. Are they well-dispersed, aggregated, or oriented in a specific way. This is critical for electrical and mechanical properties. It cannot be identified with a FTIR.
  7. Authors should include SEM images of CNT-Ecoflex and CNT-Ecoflex in presence of IPA and how it it making difference in terms of dispersion state.
  8. Pg 12 Line 327: “The optical characteristics of the dried conductive CNT ink surface were also analyzed, revealing that an increased volume of IPA enhanced the porosity of the solidified ink, a characteristic that could be reduced by increasing the amount of MWCNTs.” Authors should discuss in detail the optical characterization results and how it shows increase in porosity. Quantify this porosity.
  9. Pg 12: “According to the FTIR analysis, Ecoflex, its curing agent, IPA, and MWCNTs were likely physically mixed without chemical reorganization, implying that each element can independently contribute distinct material characteristics to the conductive CNT ink.” Literature suggests that functionalized CNTs form stable interface with silicon based polymers, which enhanced load transfer during stretching cycling (Ref. Chemically and mechanically robust SWCNT based strain sensor with monotonous piezoresistive response for infrastructure monitoring). Authors should explain how non-functionalized CNT and Ecoflex make compatible interface for enhanced load transfer.
  10. Authors should discuss the effect of content of MWCNT on the electrical conductivity and display the variation of log conductivity vs log (m-mc) with mc being the percolation threshold.
  11. Literature suggest when utilized CNT-silicon polymer composite, it demonstrates strain tolerance with non-monotonous resistance change in response to applied strain. Compos. Part A Appl. Sci. Manuf. 101 (2017) pp 41-49; J. Mater. Chem. C 4 (2016) pp 157-166; Sensors Actuators, A Phys. 179 (2012) pp 83-91;  ACS Appl. Mater. Interfaces 4 (2012) pp 3508-3516. They reported the shoulder peak as second peak and the first peak corresponds to maximum strain applied to the sensor. Not only with carbon nanotubes, this behavior is also demonstrated by graphene nanofillers in TPU also. Authors should explain this in detail.
  12. Authors should provide cycling stability performance of the MWCNT-Ecoflex.

Author Response

Reviewer #3: The article titled “Highly Dispersible and Stable Carbon Nanotube Ink with Silicone Elastomer for Piezoresistive Sensing” by Lee et al. discuss development of conductive ink based on carbon nanotubes (CNTs), platinum-catalyzed silicone elastomer and isopropyl alcohol (IPA). The work presents a potentially high-performing material, and authors claims are exceptional. However, there are few section which may be improved for making this work more comprehensive. Below are the specific points that need attention:

1) Pg2 Line 44: “For instance, CNTs can be mixed with various solvents to achieve homogeneous dispersion of particles on thin films or targeted areas [9,10]. However, the dispersibility of CNTs is limited in certain solvents, often restricting the concentration of CNTs in the solution.” These two statements are contradictory. Kindly improve this.

Response: The authors appreciate the reviewer’s comment. The first sentence was to mention that CNT particles can be better dispersed with the aid of certain solvents for thin film-related applications. The second sentence was to mention that there is a limit in terms of concentration because the solvents utilized for the dispersion of CNTs are adapted only to lower concentrations. Therefore, the authors believe that these two sentences are not contradictory. However, for the better understanding from the future readers, the authors improved this part in the revised manuscript [page #2, 1st paragraph] as follows,

For instance, CNTs can be mixed with various solvents to achieve homogeneous dispersion of particles on thin films or targeted areas [9,10]. However, the dispersibility of CNTs is limited in the solvents, often restricting the concentration of CNTs in the solution because the excessive amount of CNTs cannot be easily dispersed.

2) Pg2 Line 64: “Similar to silicone elastomer-based conductive polymers, several types of conductive inks have also been reported, where key factors include, with key attributes being electrical conductivity, adhesion, flexibility, stability, viscosity, cost, and safety.” The language may be improved here.

Response: The authors appreciate the reviewer’s comment. The authors modified this sentence as the reviewer’s suggestion [page #2, 2nd paragraph] as follows,

Similar to silicone elastomer-based conductive polymers, various types of conductive inks have also been reported, where key factors were analyzed such as electrical conductivity, adhesion, flexibility, stability, viscosity, cost, and safety.

3) Pg3 section 2.2. Preparation of the conductive CNT ink: in accordance with literature MWCNT remains in bundle form and requires ultrasonication to disperse with any silicon based polymer. Authors should explain how simple manual mixing and stirring formed uniform dispersion of CNT and Ecoflex?

Response: The authors appreciate the reviewer’s comment. Usually, the ultrasonication is recommended to effectively disperse CNTs in the solution. However, the authors found that the heating occurring during the ultrasonication could affect the accelerated solidification of Ecoflex. For this reason, the authors decided to utilize a blender to stir the uncured Ecoflex and the mixture of IPA and MWCNTs. For the analysis of the effective dispersion of CNTs in the presence of Ecoflex and IPA, the electrical resistance was measured in Figure 5 and Figure 6, implying that the solidified CNT particles encapsulated in the cured Ecoflex substrate were well dispersed without causing particular irregularities. Therefore, it is assumed that the manual mixing and stirring methods can be an alternative option for effective dispersion. The authors added supplementary information in the section 2.2. of the revised manuscript [page #3, 3rd paragraph] as follows,

The uncured Ecoflex was then poured into the mixture of IPA and MWCNTs, subsequently mixed mechanically using a blender for 10 min at room temperature to avoid the unexpectedly accelerated solidification of Ecoflex caused by heat-releasing methods such as ultrasonication.

4) Authors should explain, if mixing of CNT is simple, why most literature suggest functionalization, or oxidation of CNTs before making its dispersion with any silicone based polymer?

Response: The authors appreciate the reviewer’s comment. The objective of this study was to suggest an alternative conductive CNT ink made with a facile fabrication process and a simplified chemical composition. After conducting a series of characterizations, it was found that this conductive CNT ink could be utilized for piezoresistive sensing with additional elaborations. It is also assumed that the chemical composition suggested in this study has never been reported in other studies although it shares partially same elements with some previous studies. However, the authors admit that this conductive CNT ink needs to be improved for further applications such as direct ink writing. The authors added this discussion in the revised manuscript [page #10, 1st paragraph] as follows,

This result also implies that CNTs without functionalization can be utilized for piezoresistive sensing with additional elaborations.

5) Pg5 Line 178: “When no IPA was added to the samples containing 0.1 g and 0.2 g MWCNTs (concentrations of approximately 15.7 and 27.2 wt%, respectively)”. Kinldy discuss how wt% is derived here, is it w.r.t. IPA or Ecoflex or combined?

Response: The authors appreciate the reviewer’s comment. Initially, the authors fixed the amounts of IPA and Ecoflex at 5 mL, respectively. Then, to facilitate the preparation of MWCNTs, the authors utilized MWCNTs in the range of 0.1 and 0.4 g. Using the densities of IPA and Ecoflex (0.785 g/mL and 1.07 g/mL, respectively), the authors calculated the concentration of MWCNTs in each sample. Fortunately, the authors found that there was a mistake in the calculation (adding 1 to 4 g MWCNTs instead of 0.1 to 0.4 g). The authors corrected the wt% of all the samples tested in the revised manuscript.

6) Pg6 Line 185: “It is worth noting that if the concentration of MWCNTs is not sufficiently high, the hardened conductive CNT ink will not exhibit high electrical conductivity. However, an excessively high concentration of MWCNTs would render the fluid highly viscous, eventually leading to the transformation of the conductive CNT ink into a powder-like state. Therefore, the addition of IPA is a useful method to achieve an adjusted fluidic state of the conductive CNT ink and to ensure reliable electrical conduc-tivity.” Kindly provide concentration dependent electrical conductivity data to support this claim.

Response: The authors appreciate the reviewer’s comment. To support these sentences, the authors conducted an electrical resistance measurement using 5 distinct samples. In Figure 5, the electrical resistance measured in the sample containing 0.1 g MWCNTs without IPA (1.8 wt%) was approximately 9.2 MΩ, which was high enough not to be considered as a conductive material. The electrical resistance of the sample containing 0.2 g MWCNTs without IPA (3.6 wt%) was approximately 8.1 MΩ. However, wide variations observed in this sample may be potentially related to the uneven distribution of CNTs probably caused by the increased concentration of CNTs with respect to the sample containing 0.1 g MWCNTs, which could also affect the reliability of the solidified ink for electrical resistance measurement. On the other hand, the addition of IPA in the sample containing 0.2 g MWCNTs (2.1 wt%) led to the decrease of the electrical resistance to approximately to 3 MΩ and to the diminution of variations in electrical resistance, which proved the effect of IPA. The authors added this discussion in the revised manuscript [page #6, 1st paragraph, page #9] as follows,

The relationship between the addition of IPA and electrical conductivity was explained in Figure 5b-ii.

The second sample exhibited high irregularities in electric resistance, which might have been caused by the uneven distribution of CNT particles due to its increased viscosity compared to the first sample, which was similar to a powder-like state.

7) Pg6 Figure 2b and c, do not provide any significant information. SEM image is not clear to distinguish between CNT and Ecoflex. The elemental analysis graph is showing largest peak for Si, still it has a wt conc. of 28.18.

Kindly explain this and use another SEM distinguishing CNT and polymer. It cannot tell you if the CNTs are well-dispersed, aggregated, or oriented in a specific way. This is critical for electrical and mechanical properties. A poorly dispersed sample would have terrible properties.

Response: The authors appreciate the reviewer’s comment. For the elemental analysis, it is assumed that the relationship between the number of counts, atomic concentration, and weight concentration is related to a technical factor of the energy dispersive spectrometer and the atomic weight of each element. It is known that heavier elements are easier to detect than lighter ones. In this case, Si is heavier than O and C. That would be the reason for which Si showed the largest peak, whereas the weight concentration was not proportional to the number of peak counts. The objective of the SEM analysis was to understand the dispersing behavior of MWCNTs in the cured Ecoflex. However, given that not every MWCNT particle was on the surface, it was somewhat complicated to analyze the dispersion of MWCNTs with the more detailed SEM analysis. However, the authors believe that the optical images of the conductive CNT ink samples would be able to help the future readers understand the approximative patterns of the differently prepared conductive CNT inks. The authors added this discussion in the revised manuscript [page #6] as follows,

Compared to O and C, Si showed the highest peak, which is possibly related to a technical factor of the equipment in combination with its atomic weight. Another thing to note is that given that not every MWCNT particle was on the surface, the analysis was limited because Ecoflex is not conductive.

8) Pg 7 Figure 3: Authors should add FTIR of MWCNT in comparison with other 4 spectra. Authors should discuss in detail about specific peaks from the Ecoflex (like Si-O-Si, Si-CH3) or IPA (O-H) monitored for significant shifts or shape changes that would indicate even weak interactions like hydrogen bonding between SWCNT and polymer?

Response: The authors appreciate the reviewer’s comment. The authors think that the interaction between MWCNT and the elements can be analyzed using Figure 3d. When it comes to comparison between Figure 3c and Figure 3d, it is assumed that no particular peak or shift was made in a significant way after the addition of MWCNT. For the peaks from IPA, some characteristic peaks such as O-H band were found between 3000 and 4000 cm-1, while for the peaks from Ecoflex, some peaks such as Si-O-Si and Si-CH3 bands were found between 1000 and 1100 cm-1 and in 2960 cm-1, respectively. However, for the peaks from the mixture of IPA and Ecoflex, the common peaks such as O-H and Si-O-Si bands were found, implying the potential physical mixture of each component. When it comes to hydrogen bonding, the authors believe that it would be difficult for non-functionalized MWCNT to create hydrogen bonding with Ecoflex because both elements do not have functional groups that can help the formation of hydrogen bonding. The authors added this discussion in the revised manuscript [page #7, 1st paragraph] as follows,

In other words, the O-H band from IPA located between 3000 and 4000 cm-1, the Si-O-SI and Si-CH3 bands from Ecoflex located between 1000 and 1100 cm-1, and in 2960 cm-1, respectively, remained in the mixture of IPA and Ecoflex.

9) Authors have identified modifications in light absorbance (Line 204). Kindly quantify these changes (like, peak shifts in wavenumbers, percentage change in intensity) and discuss their potential physical origin, such as hydrogen bonding or scattering effects from the MWCNT.

Response: The authors appreciate the reviewer’s comment. As far as the authors know, the FTIR analysis of a mixture consisting of Ecoflex, IPA, and MWCNTs would be the first to report. For the slight modifications mentioned in this study, the authors assume that there could be several factors affecting these modifications such as concentration changes of each element, peak overlaps, partial heterogeneity. Further discussions could be made in the continuous study. The authors added this discussion in the revised manuscript [page #7, 1st paragraph] as follows,

Regarding modifications in light absorbance, it is assumed that several factors such as concentration changes and peak overlaps could affect this result.

10) Authors should discuss how IPA + Ecoflex sample for FTIR prepared. Since IPA is a volatile solvent and Ecoflex cures, was this spectrum taken from a liquid mixture, a dried film, or a cured elastomer? Please clarify the sample state for all spectra, as this have a major affect on the interpretation.

Response: The authors appreciate the reviewer’s comment. Every sample in the FTIR analysis was utilized right after being prepared as liquid mixtures to avoid the issues mentioned by the reviewer. The authors think that if the samples are used for the same analysis after being kept open for a certain period, they would start to solidify with the evaporation of IPA and the curing of Ecoflex. In this case, there could be unexpected and irregular variations in spectra. For example, the peaks from IPA could be attenuated and those from Ecoflex could appear differently due to the crosslinking between Ecoflex and its curing agent. The authors added this information in the revised manuscript [page #4, 1st paragraph] as follows,

Fresh samples with different compositions were prepared (IPA only, Ecoflex only, IPA-Ecoflex composite, IPA-Ecoflex-MWCNT composite) as liquid mixtures to avoid irregularities caused by the evaporation of IPA and the solidification of Ecoflex.

11) Mechanical properties of CNT Ecoflex depend on the dispersion and interface between the CNTs and the Ecoflex. Authors should provide data microscopy (SEM/TEM) or Raman mapping to corroborate the physical state of dispersion and quantify the bundle size of CNT.

Response: The authors appreciate the reviewer’s comment. As the reviewer mentioned, mechanical properties could be affected by the interaction between MWCNT particles and Ecoflex. However, the mechanical analysis in Figure 4 demonstrated that MWCNT particles were well dispersed in the cured Ecoflex. Given that the objective of this study was to show that the conductive CNT ink is the first to report with a simplified chemical composition and a facile fabrication process, it is assumed that the quantitative analysis including the bundle size of CNT in combination with SEM/TEM or Raman spectroscopy could be discussed in the continuous study.

12) Authors should propose the mechanism for the electrical and mechanical properties in CNT Ecoflex. Like, is the electrical conductivity achieved through a percolating network of CNTs within the Ecoflex matrix, and how does the IPA facilitate its formation before evaporating?

Response: The authors appreciate the reviewer’s comment. According to the results obtained in this study, the authors assume that the percolating network of CNTs and the molecular distance between each particle within the Ecoflex matrix creates electrical conductivity, which was also proved in Figure 5 with different percentages of MWCNTs in each sample. A similar explanation was also found in the reference #16 in the revised manuscript [page #14]. Regarding the function of IPA, especially in Figure 5b-ii, it was found that the sample containing 0.2 g MWCNTs with IPA shows better material properties than that containing 0.2 MWCNTs without IPA in terms of electrical conductivity and stability. This result implies that the addition of IPA in the mixture of MWCNTs and Ecoflex facilitates the dispersion of MWCNTs before the evaporation of IPA. The authors added this discussion in the revised manuscript [page #8, 2nd paragraph] as follows,

It is assumed that electrical conductivity is achieved through a percolating network formed by MWCNTs and Ecoflex.

13) Authors should demonstrate how morphology of CNTs changes with multiple stretching cycles. Are they well-dispersed, aggregated, or oriented in a specific way. This is critical for electrical and mechanical properties. It cannot be identified with a FTIR.

Response: The authors appreciate the reviewer’s comment. In this study, the conductive CNT ink was suggested with a simplified chemical composition and a facile fabrication process for piezoresistive sensing. For this reason, the authors mainly focused on electrical properties related to piezoresistive characteristics. The goal of FTIR was to clarify the possible interaction between each element, which revealed that no significant chemical reaction was verified enough to induce significant peak shifts. The authors also believe that especially in the sample containing 0.3 g MWCNTs with IPA, the effective dispersion of MWCNTs was confirmed as shown in Figure 5b-ii and also in Figure 6, showing repetitive electrical resistance values. The additional information such as the orientation of MWCNT particles could be discussed in the continuous study.

14) Authors should include SEM images of CNT-Ecoflex and CNT-Ecoflex in presence of IPA and how it it making difference in terms of dispersion state.

Response: The authors appreciate the reviewer’s comment. In this study, for the SEM analysis, the authors utilized hardened samples owing to technical problems related to the SEM equipment. In addition, as long as IPA remains in the mixture of MWCNTs and Ecoflex, the mixture would not solidify completely, which makes it difficult to analyze the samples with SEM. For this reason, the authors analyzed differently the samples containing IPA at the beginning and those without IPA from the beginning using electrical resistance measurement in a repetitive manner, showing the reduced variations in electrical resistance compared to the samples without IPA from the beginning.

15) Pg 12 Line 327: “The optical characteristics of the dried conductive CNT ink surface were also analyzed, revealing that an increased volume of IPA enhanced the porosity of the solidified ink, a characteristic that could be reduced by increasing the amount of MWCNTs.” Authors should discuss in detail the optical characterization results and how it shows increase in porosity. Quantify this porosity.

Response: The authors appreciate the reviewer’s comment. When it comes to Figure 2a, it was found that compared to the sample containing 0.2 g MWCNTs without IPA, the sample containing 0.2 g MWCNTs with IPA showed an increased porosity from the optical images. This result is supposed to be related to the difference in concentration of MWCNTs. In other words, the concentration of MWCNTs in the sample containing 0.2 g MWCNTs with IPA is approximately 2.1 wt%, whereas that in the sample containing 0.2 g MWCNTs without IPA is approximately 3.6 wt%. The authors assume that the variations in concentration of MWNCTs would be able to explain the difference in porosity of the solidified ink, which is related to the quantification of porosity in each sample. However, the porosity of the solidified ink could be also affected by several factors such as dispersion, aggregation, the way of solidification, etc. Therefore, the additional analyses including the quantification of porosity in the solidified ink could be discussed in the continuous study. The authors added this discussion in the revised manuscript [page #13] as follows,

However, the porosity of the solidified ink could also be affected by several factors such as dispersion, aggregation, and the way of solidification.

16) Pg 12: “According to the FTIR analysis, Ecoflex, its curing agent, IPA, and MWCNTs were likely physically mixed without chemical reorganization, implying that each element can independently contribute distinct material characteristics to the conductive CNT ink.” Literature suggests that functionalized CNTs form stable interface with silicon based polymers, which enhanced load transfer during stretching cycling (Ref. Chemically and mechanically robust SWCNT based strain sensor with monotonous piezoresistive response for infrastructure monitoring). Authors should explain how non-functionalized CNT and Ecoflex make compatible interface for enhanced load transfer.

Response: The authors appreciate the reviewer’s comment. It is assumed that the reference mentioned by the reviewer talks about a strain sensor based on oxidized SWCNT and polydimethylsiloxane (PDMS) for infrastructure monitoring. The authors would like to mention that the objective of this study is the suggestion of a conductive CNT ink with an easy fabrication process and a simplified chemical composition, which is adapted to stencil printing and potentially to direct ink writing. From this point of view, the authors think that the hardened conductive CNT ink realized with pristine MWCNTs showed reliable performances in terms of mechanical and electrical properties as shown in the revised manuscript. For this reason, the authors believe that it would be difficult to directly compare these two studies in the same way.

17) Authors should discuss the effect of content of MWCNT on the electrical conductivity and display the variation of log conductivity vs log (m-mc) with mc being the percolation threshold.

Response: The authors appreciate the reviewer’s comment. In this study, the samples containing MWCNTs with different concentrations with and without the presence of IPA in the form of ink were utilized for the analysis of electrical conductivity. And in Figure 6, the changes in electrical resistance under various compressive forces were found to be reliable up to the set force of 0.2 N. In addition, it is assumed that the logarithmic plot can be used to explain the dimensionality of a conductive network. However, given that the essential idea of this study is the suggestion of a conductive CNT ink rather than the development of a piezoresistive sensor, the authors think that additional analyses including the logarithmic plot of electrical conductivity could be added in the continuous study for the elaboration of a piezoresistive sensor made with the same conductive CNT ink.  

18) Literature suggest when utilized CNT-silicon polymer composite, it demonstrates strain tolerance with non-monotonous resistance change in response to applied strain. Compos. Part A Appl. Sci. Manuf. 101 (2017) pp 41-49; J. Mater. Chem. C 4 (2016) pp 157-166; Sensors Actuators, A Phys. 179 (2012) pp 83-91;  ACS Appl. Mater. Interfaces 4 (2012) pp 3508-3516. They reported the shoulder peak as second peak and the first peak corresponds to maximum strain applied to the sensor. Not only with carbon nanotubes, this behavior is also demonstrated by graphene nanofillers in TPU also. Authors should explain this in detail.

Response: The authors appreciate the reviewer’s comment. The authors understand the reviewer’s point of view. However, this study is mainly focused on a series of analyses for a conductive CNT ink rather than for strain sensors. In addition, piezoresistive sensing using the solidified conductive CNT ink was to show one of the applications of this ink. Therefore, the authors believe that the issues raised by the reviewer could be discussed in the continuous study.

19) Authors should provide cycling stability performance of the MWCNT-Ecoflex.

Response: The authors appreciate the reviewer’s comment. The material characteristics that affect the cycling stability performance of the hardened MWCNT-Ecoflex could be decided by several factors such as the concentration of MWCNTs, the presence of IPA, the way of solidification including temperature, etc. However, in this study, the authors firstly intended to mention that the newly suggest conductive CNT ink has huge potential in terms of a facile fabrication process and a chemically simplified composition, which was explained in the revised manuscript. Therefore, the authors think that the analysis concerning cycling stability performance could be raised in the continuous study.

Round 2

Reviewer 3 Report

Comments and Suggestions for Authors

The authors' efforts in revising the manuscript are highly appreciated. The revisions have significantly improved the manuscript, and it is now acceptable for publication.